# Understanding Rotavirus Vaccine Efficacy and Effectiveness in Countries with High Child Mortality

**DOI:** 10.3390/vaccines10030346

**Published:** 2022-02-23

**Authors:** Tintu Varghese, Gagandeep Kang, Andrew Duncan Steele

**Affiliations:** 1The Wellcome Trust Research Laboratory, Division of Gastrointestinal Sciences, Christian Medical College, Vellore 632004, India; tintu.varghese@cmcvellore.ac.in (T.V.); gkang@cmcvellore.ac.in (G.K.); 2Enteric and Diarrheal Disease, Bill & Melinda Gates Foundation, Seattle, WA 98102, USA

**Keywords:** rotavirus, rotavirus vaccines, acute gastroenteritis, vaccine efficacy, vaccine effectiveness, indirect effects

## Abstract

Rotavirus claims thousands of lives of children globally every year with a disproportionately high burden in low- and lower-middle income countries where access to health care is limited. Oral, live-attenuated rotavirus vaccines have been evaluated in multiple settings in both low- and high-income populations and have been shown to be safe and efficacious. However, the vaccine efficacy observed in low-income settings with high rotavirus and diarrheal mortality was significantly lower than that seen in high-income populations where rotavirus mortality is less common. Rotavirus vaccines have been introduced and rolled out in more than 112 countries, providing the opportunity to assess effectiveness of the vaccines in these different settings. We provide an overview of the efficacy, effectiveness, and impact of rotavirus vaccines, focusing on high-mortality settings and identify the knowledge gaps for future research. Despite lower efficacy, rotavirus vaccines substantially reduce diarrheal disease and mortality and are cost-effective in countries with high burden. Continued evaluation of the effectiveness, impact, and cost–benefit of rotavirus vaccines, especially the new candidates that have been recently approved for global use, is a key factor for new vaccine introductions in countries, or for a switch of vaccine product in countries with limited resources.

## 1. Introduction

Global surveillance data show that nearly 40% of diarrheal hospitalizations among children under 5 years of age are rotavirus-related and that diarrhea continues to be a leading cause of mortality in children [1,2]. Children of low- and lower-middle income countries (LMICs) are at highest risk with lack of access to safe water and improved sanitation, and urgent medical care leading to high diarrheal-associated mortality [2]. The Democratic Republic of Congo, Nigeria, Angola, India, and Pakistan together account for more than half of all rotavirus deaths worldwide [2]. Although sanitation and hygiene contribute to diarrheal disease prevention [3], the ubiquitous nature of rotavirus infection, explosive nature of clinical rotavirus disease, and the young age of infected infants makes rotavirus particularly deadly. For rotavirus infection, vaccination has been considered a critical control strategy for the past 40 years. 

Within 10 years of the discovery of rotavirus by Ruth Bishop and colleagues in Melbourne, Australia in 1973 [4] and the following widespread description of rotavirus globally, the first live-attenuated rotavirus vaccines were being evaluated in human subjects. Several lessons arose from these early clinical trials, demonstrating that the vaccines (i) worked better in high-income countries (HICs) with low rotavirus mortality [5], (ii) were more effective against severe clinical disease, (iii) that multiple doses were needed, and (iv) that efficacy against heterotypic strains had to be evaluated [6]. The first licensed rotavirus vaccine, RotaShield, (Wyeth Laboratories, Inc., Marietta, PA, USA), was approved by the FDA in 1998, after confirming most of these findings in trials in the USA, Finland, and Venezuela [7]. The vaccine, developed by the US NIH, was a reassortant rhesus-rotavirus candidate expressing four of the most commonly occurring outer capsid viral proteins to affect heterotypic protection [6]. However, the vaccine was removed from the market approximately 9 months after its introduction in the USA because of an association with intussusception [8], which is now recognized as a rare side effect associated with multiple live-attenuated oral rotavirus vaccines although not at the same level of risk as that observed with RotaShield [9,10]. 

Currently, four oral rotavirus vaccines are pre-qualified by the World Health Organization (WHO) for global use, including procurement support for countries through Gavi, The Vaccine Alliance (Table 1) [11]. These products occur in different presentations with different characteristics offering countries multiple options for introduction decisions. The various presentation options should be evaluated by countries based on the cost-effectiveness of the vaccines and the programmatic constraints of each individual country. However, their vaccine efficacy should not differ and forms the basis of this discussion.

Rotarix, GSK Biologicals, Rixensart, Belgium and RotaTeq, Merck Vaccines, Whiteriver, Pennsylvania were licensed nationally in 2006, based on large efficacy and safety studies conducted in Europe, Latin America, and the United States [6,7]. Efficacy studies conducted in Africa and Asia drew attention to the discrepancies in the vaccine efficacy in countries with low childhood mortality and high child mortality settings [12,13,14,15,16]. High-mortality countries were categorized based on the WHO Under-5 childhood mortality figures by quartile [17]. Based on these clinical studies, the WHO Strategic Advisory Group of Experts (SAGE) recommended that rotavirus vaccines should be introduced in national immunization programs globally [17]. A decade later, two oral rotavirus vaccines manufactured in India, Rotavac, Bharat Biotech, Hyderabad and Rotasiil, Serum Institute, Pune were licensed in India and then pre-qualified by WHO for global use in 2018. This was extremely timely with respect to improving the global supply of rotavirus vaccines for the world.

With the addition of new vaccines to the market, it is useful to re-examine vaccination impact and cost-effectiveness data, especially in settings where the highest diarrheal mortality occurs and where the most benefit from these vaccines to prevent rotavirus-related mortality and severe morbidity is predicted. Having access to this information should be of immense value in supporting the national implementation decisions in countries yet to start vaccinating their children and in supporting choices of the individual vaccine products that best fit the country’s requirements.

## 2. Rotavirus Epidemiology

The epidemiology of rotavirus disease varies in low- and lower-middle income countries (LMICs) and high-income countries (HICs), where rotavirus mortality is low. Rotavirus diarrhea has a distinct autumn/winter seasonality in HICs, while year-round transmission and greater force of infection are described in LMICs [18]. Similarly, most cases of severe rotavirus gastroenteritis occur in younger infants and children in LMICs compared to HICs, making them more vulnerable to rapid dehydration and death. A study evaluating the incidence and severity of rotavirus diarrhea in sub-Saharan Africa and Europe illustrated that incidence of infection in children under 2 years of age was earlier in Africa with peak incidence at around 5 months of age in high-mortality settings as compared to approximately 20 months of age in low-mortality settings [19,20]. Children in the high-mortality countries also experience multiple episodes of rotavirus diarrhea with documented incomplete immunity even after two to three infections (unlike that observed in low-mortality countries), as evidenced by an Indian cohort study where two and three previous infections conferred only 57% and 79% protection against subsequent severe rotavirus gastroenteritis [21]. In light of these findings, as well as the poor performance of other oral vaccines such polio and cholera vaccines in resource constrained settings, rotavirus vaccines were expected to have lower efficacy in high mortality settings [22], making it important to measure both vaccine efficacy and effectiveness in LICs and LMICs, as recommended by the WHO [17].

### 2.1. Rotavirus Strain Diversity

It is also important to consider rotavirus strain diversity, which may or may not have an impact on the efficacy and effectiveness of a vaccine in high-burden settings [23]. Rotaviruses are classified by the characterization of the two outer capsid viral proteins—VP7, which forms the outer capsid of the virion, and VP4, which is a spike protein enabling virus–cell attachment and entry [24]. Both viral proteins have been observed to generate neutralizing antibody responses in the host [25] and have been the focus of most vaccine efforts to date [6].

The WHO global rotavirus surveillance network and several regional networks have documented the broad diversity of strains in LICs and LMICs compared to HICs [2]. Several reviews of global strains have identified the circulation of unusual rotavirus strains, such as the VP4 P [6] strains in Africa and South Asia [26,27,28]. In addition, rotavirus reassortant strains after potential zoonotic infection and reassortment events have been frequently observed in Africa and Asia and may provide challenges to the efficacy of the vaccines developed against the more commonly identified strains circulating in humans [27].

### 2.2. Efficacy Trials of Rotavirus Vaccines

Individually randomized, placebo-controlled clinical trials evaluating the efficacy and safety of all four licensed rotavirus vaccines have been reported from countries with high under-five mortality rates [9,29]. The primary outcomes evaluated in these trials were reduction in severe rotavirus gastroenteritis (SRVGE), rotavirus gastroenteritis of any severity (RVGE), and rotavirus diarrhea requiring hospitalizations in the vaccinated arm compared to the unvaccinated arm, and utilized similar protocol designs, assessment tools, and endpoints [30]. Reduction in severe diarrhea due to any cause, diarrhea of any severity, diarrheal hospitalizations, and diarrheal-associated deaths were also measured. These studies are summarized in Table 2.

A systematic review on the Rotarix and RotaTeq efficacy trials reported higher vaccine efficacy in HICs with low diarrheal mortality in comparison with high-mortality strata countries [41]. No data are available for Rotavac and Rotasiil from low-mortality settings for a similar comparison, as the clinical studies were only conducted in India and Africa [38,39,40]. 

Lower efficacy in LMICs is likely to be multifactorial due to differences in rotavirus epidemiology with the high force of infection, co-infections with other enteric pathogens, malnutrition, environmental enteropathy, interference of vaccine uptake by maternal antibodies, and co-administration of other vaccines [22,42]. Several studies have been conducted to assess various options to improve vaccine performance, measured as serum IgA immune response rather than clinical outcome, which were summarized recently [42]. These included host factors, such as prenatal macro- and micro-nutrient supplementation to improve an infant’s nutritional status, withholding breastfeeding at the time of vaccination, and zinc-probiotic supplementation; no, or only marginal, differences in seroconversion were recorded [42,43,44,45,46]. Other approaches to improve vaccine performance in high-transmission settings included alternative delivery mechanisms and schedules, such as a booster dose at 9 months of age, a delayed schedule of immunizations or neonatal dosing schedule, and two versus three doses of the Rotarix vaccine [34,37,47,48].

Despite their modest efficacy, the number of severe rotavirus gastroenteritis episodes and diarrheal deaths prevented with these vaccines can have greater impact in areas with high baseline disease. For instance, the number of severe rotavirus gastroenteritis episodes averted in the first year of life was greater in Malawi (8.4 episodes prevented per 100 children vaccinated) than in South Africa (2.5 episodes prevented per 100 children vaccinated), even though the efficacy estimates were lower in Malawi (49.4%; 95% CI: 19.2–68.3) in comparison to South Africa (76.9%; 95% CI: 56.0–88.4) [15]. This emphasizes the importance of evaluating vaccine effectiveness studies in real-world settings compared to the point estimates of efficacy from controlled, double-blind studies.

Nevertheless, although head-to-head comparisons of the different oral rotavirus vaccines have not been conducted, all four vaccines have comparable vaccine efficacy against SRVGE in infants in high-mortality countries ranging from 48% to 58% [9]. Lower vaccine efficacy was noticed in the second year of life in high-mortality settings in contrast to low-mortality settings where protection extended to older age [41]. In high-mortality settings, Rotarix prevented 58% of SRVGE when children were followed up for one year after vaccination, while a 35% reduction in SRVGE was noted for a two-year follow-up period [41]. Corresponding figures for RotaTeq were 57% and 44%, respectively [41]. The Rotavac phase 3 trial conducted in India showed an efficacy of 57% and 54% during the one-year and two-year follow-up periods, respectively [38]. Similarly, in India, Rotasiil prevented 48% and 44% of SRVGE during the one-year and two-year follow-up periods, respectively [40,41], and it prevented ≈28% of cases of rotavirus diarrhea with efficacy increasing with the severity of the diarrheal episode (39% against SRVGE) during one year follow-up [40].

Interestingly, the corresponding figures for Rotasiil in Niger were 60% efficacy for the first year of life and 54% for 2-years of follow-up [39]. In this setting with a high force of infection, this could be partially explained by earlier exposure to natural rotavirus infection, which would confer natural protection in the placebo group, causing a lower incidence and lower efficacy beyond the first year of life. Furthermore, the immunized cohort was infected within a few months of immunization due to the abrupt rotavirus season in Niger, and this may have artificially heightened efficacy due to this proximity to the vaccine administration [49]. Some other studies also reported the decreased rotavirus vaccine performance among children born during rotavirus season or immediately after, when they received their first dose of vaccine with high maternal antibody levels at that time [50,51]. Another reason could be waning immunity, which could be counteracted by giving booster doses [42]. It is estimated that an additional 3–16% reduction in rotavirus deaths can be achieved if vaccine efficacy is re-established by a booster dose prior to the second year of life [52]. Even without booster doses, rotavirus vaccines will have a large impact, as the disease burden and risk of deaths is highest during the first year of life in high-mortality settings. Data show that >75% of African children are exposed to wild-type virus by their first birthday and only 10% of children above one year required hospitalizations for rotavirus illness [53,54].

Nearly 27% of under-five deaths due to all causes were averted with rotavirus vaccines in the high-mortality settings [41]. A recent meta-analysis of efficacy data revealed nearly a 16% reduction in severe all-cause diarrhea with Rotarix, RotaTeq, and Rotavac at two years follow-up [41]. Rotarix efficacy was 25.1% (95% CI: 4.7–40.8) against severe diarrhea of any cause in a study conducted in Malawian and South African infants during a one-year follow-up period [35]. In the high HIV prevalence population of Kenya, RotaTeq efficacy was 34.4% (95% CI: 5.3–54.6) against severe all-cause diarrhea in the first follow-up period [33].

### 2.3. Rotavirus Vaccine Introduction Status

The WHO recommended universal rotavirus vaccination in 2009, and by November 2021, 114 countries had introduced the rotavirus vaccine [55] (Figure 1). Asian nations lag behind in vaccine implementation, while greater than 70% of Sub-Saharan African countries have introduced rotavirus vaccines. A major contributor to the introduction of vaccines in most African countries is Gavi, The Vaccine Alliance, which provides vaccine subsidy support for eligible countries, and this may explain some of the hesitation in Asia. Many countries in the region are not Gavi-eligible or transitioning from Gavi financing toward self-financing [56]. Lack of awareness about rotavirus disease estimates, worries about vaccine safety, high vaccine costs, lack of political will, challenges with programmatic adoption, and concerns about the sustainability of vaccinations are the main obstacles to vaccine introduction [57].

Several high-mortality countries have not begun immunizing their children yet, and globally, nearly 55 million children lack access to vaccines [58]. The Democratic Republic of Congo, a Gavi-eligible country with one of the world’s highest rotavirus mortality rates, introduced the rotavirus vaccine in December 2019, and Nigeria, with the highest rotavirus mortality in sub-Saharan Africa, was anticipated to introduce the rotavirus vaccine in 2020, which was inevitably delayed by the COVID-19 pandemic. They are slated to introduce the vaccine, with Gavi support, in 2022 [58]. Experience shows that the rotavirus vaccine introduction provides a great opportunity to strengthen the routine immunization program by identifying the gaps for each component of the Universal Immunization Program (UIP) such as rectifying cold chain constraints, staff shortages, improving accessibility to outreach areas, rationalization of available inventory, and procurement of necessary equipment [59]. Yet there remain countries that will need to be convinced of the benefits of rotavirus vaccines by assessing vaccine effectiveness, costs, and impact data.

### 2.4. Impact and Effectiveness of Rotavirus Vaccines

A post-licensure evaluation of vaccine effectiveness (VE) represents how well a vaccine performs in real-world settings, which helps policymakers confirm benefits, plan immunization schedules, and guide future vaccine development and immunization strategies. The majority of the early rotavirus VE studies were conducted in low-mortality settings as early as 2006, while those from high-mortality settings were reported mostly after 2016 and when several countries had introduced (Table 3). 

Most studies utilized a case-control design to calculate the VE assuming that the unvaccinated and vaccinated children have a comparable risk of developing the disease. Cases are children with laboratory-confirmed rotavirus diarrhea presenting to a health care facility or hospital, while control groups are either test negative-diarrheal controls (children admitted with non-rotavirus diarrhea) or non-diarrheal controls (children admitted to hospital for non-diarrheal diseases or community controls). VE estimation using the test-negative diarrheal controls better addresses potential confounders related to health care-seeking behaviors and generally match well for socioeconomic status and pathogen exposure. The findings discussed below are mainly based on Rotarix and RotaTeq studies, as the post-licensure data are not yet available for the newly licensed vaccines.

Meta-analysis of VE data from 2006 to 2019 found an overall VE against rotavirus diarrheal hospitalizations to be 58% for Rotarix and 45% for RotaTeq in high-mortality countries, which is much less than in low-mortality countries (83% and 85%, respectively) [73]. Countries also witnessed a reduction in all-cause diarrheal hospitalizations and diarrheal-related deaths after vaccine introduction. In just one year after the introduction of the vaccine, Nicaragua reported a significant decrease in diarrhea-associated outpatient visits, hospitalizations, and infant mortality (27%, 12%, and 30%, respectively) [74]. Based on a time-series analysis from Rwanda, diarrheal hospitalizations dropped by 29% in the first year following vaccine introduction, rising to 47% when analyzing data only for the rotavirus season [75]. Globally, the diarrheal deaths decreased by 24% from 2.18 million deaths in 2006 to 1.66 million deaths in 2016, a decade after rotavirus vaccine introduction [76]. Finally, it was estimated that if all Asian countries introduced vaccines, nearly 49% of hospitalizations and 40% of deaths can be further prevented [77]. Similarly, in Africa, an estimated reduction of 47% in hospitalizations and 39% of deaths can be achieved with full vaccine coverage across the continent including DRC and Nigeria [78].

The vaccine impact will be dependent on the effectiveness of the vaccine, the timeliness of the completion of the schedule, and the immunization coverage. A considerable level of protection is provided by partial vaccination series, which is reassuring for the children of LMICs who often fail to complete the series or complete the series in a delayed time frame when rotavirus exposure is also high [79]. However, while there is some benefit, the protection from a partial vaccination is never as good as that from a completed schedule, underlining the importance of complete and timely immunization for all children [79,80]. For instance, in Malawi, the concomitant reduction in the incidence of RVGE hospitalizations in infants (268.7 in 2013, 152.5 in 2014, and 123.1 in 2015 from January to June per 100,000 age-adjusted population) was demonstrated with an increase in vaccine coverage (75% in 2013, 92% in 2014, and 95% in 2015) [69]. Similarly, post-vaccination data from India revealed a greater reduction in rotavirus diarrhea from states with higher vaccine coverage [81].

Measurement of the indirect effects of rotavirus vaccines is important when evaluating the total impact of vaccination programs, particularly in LMICs with a higher disease burden and lower immunization coverage. In resource-poor settings, it is often difficult to estimate indirect effects accurately [82], and a few studies conducted in LMICs have had ambiguous results [36,75,83]. In Malawi, a reduction in rotavirus episodes was seen in unvaccinated infants, while there was no such reduction in children above 1 year [84]. A 28% reduction in rotavirus hospitalization in unvaccinated children <1 year was reported from Tanzania [70]. Rwanda reported a 31–34% reduction in diarrheal admissions among 1–4-year-old children two years after vaccine introduction [75]. A study by Otieno et al. in Kenya also suggested evidence of herd protection [85]. In India (unpublished data), a substantial drop in rotavirus hospitalization by 40–50% was observed in 2–5-year-old children with Rotavac. However, no such indirect effects were reported from Bangladesh [36].

A drop in VE in the second year of life, with a shift in rotavirus diarrhea to older age groups, was reported in a few studies [62,69]. Hence, expanding vaccine protection beyond the first year of life should be considered. Some studies have reported that the drop in efficacy in the second year of life is limited to subgroups such as the HIV-exposed and chronically malnourished children [68,69], although the numbers are fairly small in these reports. In considering ways to address these issues, a booster dose of rotavirus vaccine at 9 months along with other vaccines was explored [47,52,86,87]. A clinical trial conducted in Malian infants confirmed that the co-administration of booster doses of RotaTeq vaccine did not interfere with the seroconversion of measles and Meningitis A vaccines [86]. The booster doses were well tolerated and induced a better protective immune response in vaccinated infants. Similar findings were obtained from the Bangladesh trial when rotavirus vaccine boosters were co-administered with Measles–Rubella vaccines [47]. Enhanced immune response and prolonged protection were shown, but the potential benefits for the entire population or subsets need to be weighed against added costs before making policy recommendations for booster doses. The additional doses might be of less value in low-resource settings where the major burden is among infants [52]. Continued surveillance would provide useful data on the potentially changing rotavirus epidemiology and the need for additional vaccination.

The effectiveness of rotavirus vaccines against common circulating strains and non-vaccine strains has been monitored since vaccines differ in their composition, and because it was initially unclear whether the vaccines would provide comparable heterotypic protection, and whether strains would evolve to escape the protective immune response. Although changes in viral circulation have been reported, these have been observed as the usual natural changes occurring in rotavirus strain diversity given its segmented genome and not necessarily different in countries that did or did not introduce vaccines. Rotarix and RotaTeq have proven effective against homotypic and heterotypic rotavirus strains in efficacy and effectiveness studies [49]. According to pooled strain-specific VE, Rotarix was 59% effective against homotypic strains, 72% effective against partially heterotypic strains, and 47% effective against fully heterotypic strains. Similarly, 70% VE was achieved against homotypic strains and 87% was achieved against heterotypic strains with RotaTeq [88]. However, only a few LMICs, which classically have more genetic virus diversity, were included in this meta-analysis, and it will be worthwhile to re-examine the data periodically and particularly with the introduction of new vaccines and with longer-term vaccine usage. Currently, limited published real-world data are available for Rotavac [81] and none are available for Rotasiil. Nonetheless, a reasonable heterotypic coverage is anticipated, as the efficacy trials of these vaccines showed heterotypic protection [38,40]. Continued post-vaccine surveillance to synthesize genotype specific VE data will be helpful to confirm these findings.

In the absence of a head-head comparison of various rotavirus vaccines, existing data are insufficient to favor one product over another. The availability of multiple vaccine products creates a challenge when the same product might not be available to complete the series. The current WHO recommendation is to immunize the child with the same product whenever feasible, but the emphasis should be on completing the series [9]. Any licensed product could be utilized, even if the product used for the prior dose is not available and restarting the series is not necessary [9]. Equivalent VE was reported for completed vaccination series with a single vaccine product and mixed vaccines (three-dose regimen with mixed Rotarix and RotaTeq) from the USA [89,90]. There is a need to generate similar data for the newly approved vaccines, as this opens the possibility of modifying the existing schedule with affordable vaccines, ensuring the sustainability of vaccination programs.

Rotavirus vaccines are highly cost-effective or even cost saving in the world’s low-, lower-middle and even middle-income countries [91,92,93]. Although Gavi support has been essential for national decisions around rotavirus vaccine introduction in many countries, modeling studies indicate continued cost-effectiveness even after transitioning from Gavi support and when the country would be self-financing the immunization program [73,74,76]. The availability of newer vaccines in the global market along with transparent and flexible pricing mechanisms is expected to bring down the cost of rotavirus vaccines further. A recent benefit–cost analysis of rotavirus vaccine introduction in eight sub-Saharan African countries which are yet to introduce them [56,92,94] found that for all Gavi-eligible countries, Rotarix offered the highest cost–benefit ratio, while Rotasiil and Rotavac were the optimal choices for non-Gavi countries. Lower vaccine price per dose led the government of the Palestinian Territories to switch to Rotavac from Rotarix when vaccine subsidy support from the Rostropovich Foundation ended, and it proved to be cost saving for them [95,96]. Multiple factors beyond the vaccine price including the costs of delivery, cold chain volume, product presentation, doses per the schedule, and vaccine wastage need to be systematically assessed before making a choice, but it is clear that rotavirus vaccination is advantageous as compared to standard treatment regimes in high-mortality countries [92].

### 2.5. Ongoing Rotavirus Vaccine Research

Specifically, for LMICs with a high force of infection leading to an early disease burden, immunization schedules starting from birth are expected to have added benefits over existing schedules in providing earlier protection, the opportunity for better vaccine coverage, and a potential safety benefit, as intussusception is rare among neonates [48]. A novel oral rotavirus vaccine, RV3-BB, developed from a unique neonatal strain G3P [6] in Australia has shown promising results. A Phase 2b clinical trial in neonates in Indonesia showed comparable vaccine efficacy when it was administered in a neonatal schedule given at birth, followed by doses at 8 and 14 weeks, with the infant schedule (8, 14, and 18 weeks) [48]. These encouraging results were recently consolidated with robust immunogenicity from a study conducted in New Zealand with the same vaccine in neonates. This approach could possibly be replicated with Rotavac, which is based on a neonatal strain identified in India, if the neonatal schedule proves to be desirable by high-mortality countries.

Another focus to improve vaccine performance is the development of injectable rotavirus vaccines, based on the inactivated polio vaccine model, to address the lower efficacy of oral vaccines in LMICs. The lead candidate in this category is the trivalent, subunit vaccine P2-VP8* candidate, which was discovered at NIH and is now in clinical development by PATH, which is currently undertaking a Phase 3 efficacy trial in multiple African countries after successful Phase 1 and 2 studies [97,98]. Non-replicating formulations of already licensed oral vaccines, mRNA vaccines, virus-like particle, and nanoparticle vaccines are also in early stages of development [99]. Superior efficacy is expected with the injectable vaccines as they circumvent the purported reasons for the suboptimal efficacy of oral vaccines, which were described previously [42,100]. Other advantages include the potential increased safety profile, reduction in cold-chain footprint, lower costs, and prospect of being co-formulated with other parentally delivered vaccines. Even so, it is possible that they may be deployed to augment rather than replace the oral vaccines, similar to the polio immunization strategy.

## 3. Conclusions

Where healthcare is hard to access, prevention is the best way to protect children against diarrheal diseases. Although the efficacy of rotavirus vaccines is higher in low-mortality settings, the impact is greater in high-mortality countries because of the high disease burden. Effectiveness studies have demonstrated a reduction in diarrheal diseases and death, the protection offered even by incomplete vaccination series and heterotypic protection against diverse genotypes. Nonetheless, concerns include low vaccination coverage in some countries, the waning of protection beyond the first year of life in some subsets of children, and the sustainability of the vaccination program without donor support. With two new WHO pre-qualified vaccines now available to strengthen the supply pipelines, the continued generation of effectiveness data with these new vaccines will help ensure further vaccine introductions such that children everywhere can benefit from rotavirus vaccination.

## Figures and Tables

**Figure 1 vaccines-10-00346-f001:**
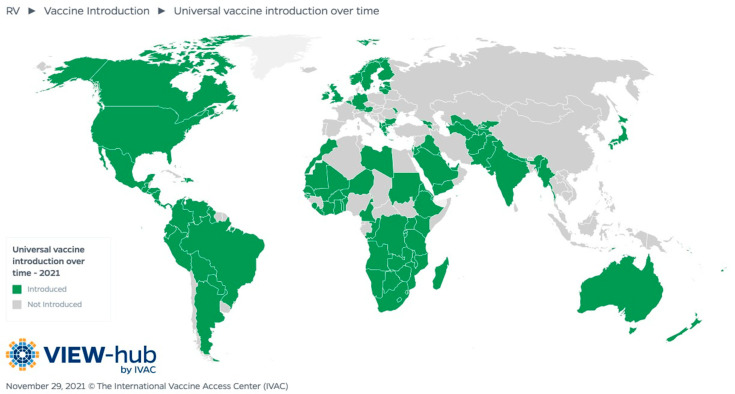
Rotavirus vaccine introduction status.

**Table 1 vaccines-10-00346-t001:** WHO-approved rotavirus vaccines, January 2022.

Vaccine	Content	Dosage and Schedule	Storage; Vaccine Vial Monitor (VVM)	Approval Date	Cost Per Dose ^a^
RotaTeq ^b^ (Merck & Co. Inc., Whiteriver, PA, USA)	Pentavalent bovine–human rotavirus reassortant with human G1, G2, G3, G4, and P [8]	Each dose (2 mL) contains 2.0–2.8 × 10^6^ IU/serotype 3 doses given 4–10 weeks apart beginning at 6–12 weeks of age; the series should be completed by the age of 32 weeks	2–8 °C, 24 months; (None)	October 2008	$3.20
Rotarix (GSK, Rixensart, Belgium)	Monovalent attenuated human strain R1X4414 of G1P [8] strain	Each dose (1.5 mL) contains 10^6^ °C CID_50_ 2 doses given 4 weeks apart beginning at 6 weeks of age; the series should be completed by the age of 24 weeks	2–8 °C, 24 months; (VVM 7)	March 2009	$2.54
Rotavac (Bharat Biotech International Ltd., Hyderabad, India)	Monovalent attenuated human neonatal 116E of G9 P [11] strain	Each dose (0.5 mL) contains a viral titer of 10^5.0^ FFU 3 doses given 4 weeks apart beginning at 6 weeks of age; the series should be completed before the age of 8 months	Available in two forms: −20 °C, 60 months; (VVM 2) Liquid: 2–8 °C, 24 months (VVM 7)	January 2018 2021	$0.85 $1.15
Rotasiil (Serum Institute of India, Pune, India)	Pentavalent bovine–human rotavirus reassortant with human G1, G2, G3, G4, and G9	Each dose (2.5 mL) contains 10^5.6^ FFU/serotype 3 doses given 4 weeks apart beginning at 6 weeks of age; the series should be completed during the first year of life	Available in 3 forms: Lyophilized: 2–8 °C, 30 months; (VVM 30) Thermostable lyophilized: 25 °C, 30 months; VVM 250+ Liquid: 2–8 °C, 24 months; (VVM 7)	September 2018 January 2020	$0.95 $1.85 $0.80

Abbreviations: IU—infectious unit; CCID_50_—median cell culture infective dose; FFU—focus-forming units. ^a^: Source: Modified from Detailed Product Profiles of WHO prequalified vaccines, Gavi Alliance as of February 2022 [11]. ^b^: RotaTeq is no longer available to the Gavi market, however it is included as many of the data inputs assessed in this review are for RotaTeq.

**Table 2 vaccines-10-00346-t002:** Summary of vaccine efficacy trials from high-mortality countries (WHO mortality stratum D and E).

Ref.	Study Period	Location	Vaccine	Sample Size	Follow-Up Duration (Months)	Efficacy ^a^ %			
						RVGE	SRVGE	All AGE	S-AGE
[31]	2007–2009	Ghana	RotaTeq	2200	21	48.8	ND	ND	25.3
[32]	2007–2009	Mali	RotaTeq	1648	21	19.2	17.6	ND	ND
[33]	2007–2009	Kenya	RotaTeq	1308	16	62.0	63.9	10.0	10.6
[34]	2005–2007	South Africa ^b^	Rotarix	833	24	52.0 ^c^	40.0 ^c^	ND	25.0 ^c^
[35]	2006–2007	Malawi ^b^	Rotarix	1194	24	ND	38.1 ^c^	ND	15.9 ^c^
[36]	2008–2011	Bangladesh	Rotarix	11,004	21	29.0	22.9	ND	ND
[37]	2011–2013	Bangladesh	Rotarix	593	12	51.0	73.5	−3.1	22.1
[38]	2011–2012	India	Rotavac	6541	24	34.6	53.6	ND	18.6
[39]	2014–2015	Niger	Rotasiil	3508	24	29.3	54.4	6.5	11.1
[40]	2014–2015	India	Rotasiil	7034	24	22.6	39.5	ND	4.6

Note: Nonrandomized trials and studies with outcomes not relevant to this review are excluded from the table. RotaTeq, Rotavac, and Rotasiil vaccines were given at 3 doses (6, 10, and 14 weeks). Rotarix was given at 2 doses (6 and 10 weeks). Severe diarrhea: Diarrheal episodes with a Vesikari score of ≥11. CI: Confidence interval, RVGE: Rotavirus gastroenteritis of any severity, SRVGE: Severe rotavirus gastroenteritis. AGE: Acute gastroenteritis of any severity, S-AGE: Severe acute gastroenteritis, ND: No data. ^a^: Per protocol efficacy. ^b^: Studies had two vaccine arms (two-dose arm with RV1 given at 10 and 14 weeks and three-dose arm with RV1 given at 6, 10, and 14 weeks). ^c^: Efficacy of overall vaccine group (2-dose and 3-dose arms together).

**Table 3 vaccines-10-00346-t003:** Rotavirus vaccine effectiveness studies conducted in countries with high-mortality stratum.

Ref.	Location	Vaccine	Vaccine Introduction	Study Period	Children Enrolled	Age Eligible Children for VE Analysis	Vaccine Effectiveness for a Completed Series against Rotaviral Admissions of Any Severity: VE% (95% CI)		
							Children < 5 Years	≤12 Months	12–23 Months
[60]	Nicaragua	RotaTeq	October 2006	July 2007–June 2010	11,573	1974	45	64	30
[61]	Bolivia	Rotarix	August 2008	March 2010–June 2011	2545	2318	69	64	72
[62]	Bolivia	Rotarix	August 2008	April 2013–March 2014	870	776	59	76	45
[63]	Guatemala	Rotarix RotaTeq	February 2010	January 2012–August 2013					
[64]	Botswana	Rotarix	July 2012	June 2013–April 2015	667	610	54	52	67
[65]	Rwanda	RotaTeq	May 2012	September 2012–May 2015	200	200	75	65	81
[66]	South Africa	Rotarix	August 2009	April 2010–October 2012	2099	1974	57	54	61
[67]	Burkina Faso	RotaTeq	October 2013	December 2013–February 2017	1043	988	35	58	19
[68]	Ghana	Rotarix	April 2012	April 2012–December 2014	1021	657	60	78	50
[69]	Malawi	Rotarix	October 2012	November 2012–June 2015	997	933	58.3	70.6	31.7
[70]	Tanzania	Rotarix	February 2013	March 2013–December 2015	2859	ND	57	56	57
[71]	Zimbabwe	Rotarix	May 2014	June 2014–December 2017	4338	3643	ND	61	−48
[72]	Kenya	Rotarix	July 2014	July 2014–December 2017	677	509	ND	67	72

VE: vaccine effectiveness; CI, confidence interval; ND, no data. VE is calculated for completes series. Only studies with test-negative case-control design are included in this table.

## Data Availability

Not applicable.

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
