# Peer review of "Understanding Rotavirus Vaccine Efficacy and Effectiveness in Countries with High Child Mortality"

_vaccines, 2022, doi:10.3390/vaccines10030346_

Round 1

Reviewer 1 Report

This is a narrative review for rotavirus vaccine in countries with high child mortality. The authors reviewed for not only efficacy and effectiveness of rotavirus vaccine, but also for epidemiology of rotavirus infection, vaccine introduction status, and impact comprehensively. I believe that this review can contribute for understanding the importance of rotavirus vaccine.

I have some comment in below.

  • I recommend the authors to modify the title, because it is not focused on only efficacy and effectiveness.
  • The authors should make the definition of high mortality country clear.
  • The authors had better to color rotavirus vaccine introduction status (figure1) according to GAVI support in detail.
  • The authors had better to show the impact of rotavirus vaccine introduction in high mortality countries in Table like Table3.

Author Response

Thank you for your comprehensive review of our manuscript. The revisions of the manuscript as recommended by the Reviewer are described below. 

1. I recommend the authors to modify the title, because it is not focused on only efficacy and effectiveness.

Response: We felt that it was important to include the epidemiology and strain diversity of rotavirus infection in various geographic settings to give the context for the differences in vaccine efficacy and vaccine effectiveness observed. Nevertheless, we have modified the title as recommended to "Understanding Rotavirus Vaccine Efficacy and Effectiveness in countries with high childhood mortality"  

2. The authors should make the definition of high mortality country clear.

Response: We have revised the text to emphasize the definition of high mortality country (lines 82-83). The categorization was based on the interquartile range of Under-5 mortality as defined by WHO and UNICEF. 

3. The authors had better to color rotavirus vaccine introduction status (figure1) according to GAVI support in detail.

Response: The Figure was taken from the IVAC Viewhub website and will require redesigning. I would prefer to keep the original Figure as it was derived from the Viewhub website to facilitate publication.    

4. The authors had better to show the impact of rotavirus vaccine introduction in high mortality countries in Table like Table3.

Response: We have revised the Table as requested. All the countries are ranked with high childhood mortality. The top 6 countries (Bolivia, Guatemala and Nicaragua; Botswana, Rwanda and South Africa) are ranked in the second highest quartile. The remaining 6 countries are ranked in the highest quartile for Under-5 mortality.  

Reviewer 2 Report

Review: vaccines-1572030

Title: Rotavirus vaccine efficacy and effectiveness in countries with high child mortality

Authors: Varghese, Kang, and Steele.

This review describes the current state of four approved vaccines against rotavirus in low-middle income countries (LMIC), which show high mortality provoked by severe rotavirus gastroenteritis. In addition, the review revises and interprets the data obtained from different metanalysis obtained from LMIC. In general, this manuscript is well written and will be of great help for rotavirus experts, guidance for the political decision regarding rotavirus vaccines, and the community in general.

Minor changes:

Table 1: In the row for Rotasiil, the meaning of 105.6 focus forming/serotype is unclear. Please can you explain?

Table 2: Why the data for Gana are in bold?

Lines 336-338: it will be better to choose between the two parenthesis sentences: unpublished data or personal communication.

Author Response

Thank you for your review and recommendations for our manuscript. Revisions requested by the Reviewer are detailed below.

  1. Table 1: In the row for Rotasiil, the meaning of 105.6 focus forming/serotype is unclear. Please can you explain?

Response:  We have revised the text to indicate that this is a viral titer of the individual serotypes included in the vaccine, similar to the data presentation of the Merck RotaTeq vaccine construct description, also in Table 1.

We also revised the figures for ‘cost per dose’ in Table 1 which we noticed were incorrect.  

Finally, I added the data for the most recently pre-qualified vaccine which was not available previously. 

2. Table 2: Why the data for Gana are in bold?

Response: I assume that the reviewer means Ghana? I could not find the bold text, but if present, it is a typographical error which can be corrected by the Editorial staff. 

3. Lines 336-338: it will be better to choose between the two parenthesis sentences: unpublished data or personal communication.

Response: We have clarified this sentence on lines 302-304. We deleted the phrase "personal communication".